# Living Proof of Activity of Extracellular Vesicles in the Central Nervous System

**DOI:** 10.3390/ijms22147294

**Published:** 2021-07-07

**Authors:** Shadi Mahjoum, David Rufino-Ramos, Luís Pereira de Almeida, Marike L. D. Broekman, Xandra O. Breakefield, Thomas S. van Solinge

**Affiliations:** 1Program in Neuroscience, Departments of Neurology and Radiology, Massachusetts General Hospital, Harvard Medical School, Boston, MA 02119, USA; SMAHJOUM@mgh.harvard.edu (S.M.); m.broekman@haaglandenmc.nl (M.L.D.B.); breakefield@hms.harvard.edu (X.O.B.); 2CNC—Center for Neuroscience and Cell Biology, University of Coimbra, Rua Larga, 3004-504 Coimbra, Portugal; rufinoramosdavid@gmail.com (D.R.-R.); luispa@cnc.uc.pt (L.P.d.A.); 3CIBB—Center for Innovative Biomedicine and Biotechnology, University of Coimbra, Rua Larga, 3004-504 Coimbra, Portugal; 4Faculty of Pharmacy, University of Coimbra, Polo das Ciências da Saúde, Azinhaga de Santa Comba, 3000-548 Coimbra, Portugal; 5Department of Neurosurgery, Leiden University Medical Center, 2333 ZA Leiden, The Netherlands; 6Department of Neurosurgery, Haaglanden Medical Center, 2512 VA The Hague, The Netherlands

**Keywords:** extracellular vesicle, central nervous system, in vivo experiments

## Abstract

The central nervous system (CNS) consists of a heterogeneous population of cells with highly specialized functions. For optimal functioning of the CNS, in disease and in health, intricate communication between these cells is vital. One important mechanism of cellular communication is the release and uptake of extracellular vesicles (EVs). EVs are membrane enclosed particles actively released by cells, containing a wide array of proteins, lipids, RNA, and DNA. These EVs can be taken up by neighboring or distant cells, and influence a wide range of processes. Due to the complexity and relative inaccessibility of the CNS, our current understanding of the role of EVs is mainly derived in vitro work. However, recently new methods and techniques have opened the ability to study the role of EVs in the CNS in vivo. In this review, we discuss the current developments in our understanding of the role of EVs in the CNS in vivo.

## 1. Introduction

The central nervous system (CNS) consists of a heterogeneous population of cells, all functioning in a distinct milieu protected by the skull and blood–brain barrier (BBB). Neurons have an intimate relationship with oligodendrocytes, astrocytes, microglia, and endothelial cells, with the need for fine-tuned, targeted communication between each of these cell types. In disease, this need for communication is further called upon for example with the influx of inflammatory cells into the brain parenchyma. Interference in cell-to-cell communication in the brain can have a role in initiating various disease processes (e.g., neurodegeneration) and can be hijacked by tumors to hinder an adequate (inflammatory/immune) response [1,2]. Understanding this mode of communication is therefore vital in furthering our knowledge of the healthy and diseased brain.

One of the methods of cell-to-cell communication is via the release and uptake of extracellular vesicles (EVs). Extracellular vesicles (EVs) are membrane enclosed structures, typically 30 to 500 nm in diameter, originating from multivesicular bodies (MVBs) or plasma membrane [3]. These particles are naturally released from cells and contain nucleic acids, peptides/proteins, lipids, and other cellular constituents [3,4]. Initially thought to function as vehicles for disposal of cellular debris, over the years a vast array of physiological and pathological processes have been uncovered in which EVs are critically involved [5]. EVs are distinctly different from synaptic vesicles, the contents of which are released by exocytosis and the role of which is strictly tied to the signaling by neurotransmitters and neuropeptides between neurons at synapses [6,7].

Traditionally a distinction was made between ‘exosomes’—smaller particles secreted via fusion of multivesicular bodies with the cell membrane; and ‘microvesicles’ or ‘ectosomes’—larger particles shed via blebbing of the cell membrane [8]. Due to previous absence of standardized nomenclature, these terms have been used virtually interchangeably in previous studies with authors using various arbitrary measures, such as size, to infer the origin of the vesicles. Currently, the MISEV2018 guidelines state that, unless stringent imaging or subcellular marker studies are done to confirm origin, particles should be described as ‘EVs’ [4]. In this paper we will therefore use the term EVs for all extracellular particles discussed, unless specific experiments were done to establish exosomal or microvesicular origin.

EVs are released by virtually every cell in the body, and the brain is no exception. EVs have been shown to have a role in neuronal activation [9], to facilitate communication between neurons, astrocytes [10], oligodendrocytes [11], and microglia [12], and are implicated in many disease processes. However, the vast majority of current available literature is based on in vitro experiments, which are unable to take into account the complex and distinct microenvironment in which cells of the central nervous system function. In recent years, advances in intravital imaging [13], EV isolation techniques [14,15,16], and various transgenic (Cre-loxP) mouse models [17] have enabled the in vivo study of EVs in the brain, expanding our knowledge of the role of endogenous EVs in this environment.

In this review, we will discuss the current available literature on the release and functioning of EVs in the brain in vivo, under normal circumstances, and in the case of neurological disease. Use of EVs as biomarkers detected in blood [18] or cerebrospinal fluid (CSF) is beyond the scope of this review and will not be discussed [19].

## 2. The Role of Endogenous EVs in the CNS and Neurodevelopment

Neurodevelopment and neuronal circuit assembly require intricate communication between neuronal precursor cells, neurons, glia, and endothelial cells. EVs have an important role in cell signaling by facilitating transfer of biological components among these and other cell types in the brain [3,20]. Prominent roles in which EVs have been implicated include synaptic signaling [21], myelin formation [22], and neural development [23]. The current evidence based on in vivo studies is however sparse. In this section we will discuss the current in vivo based literature regarding the role of EVs in neural development and functioning.

### 2.1. Neuron–Astrocyte Communication

Astrocytes are involved in regulating the internal milieu of the brain, including the BBB, protecting neurons against reactive oxygen species, responding to neuronal injury, and maintaining synaptic transmission and plasticity [24]. As a result, astrocytes are in constant communication with neurons to respond to changes in the CNS, with EVs being important mediators [24] (Figure 1A). To visualize this EV-mediated communication between neurons and astrocytes, a mouse model expressing a Cre-dependent exosome reporter (CD63-GFP^f/f^) driven by a neuron specific CaMKII promoter was generated [17]. Intravital imaging showed that CD63-GFP labeled EVs were taken up by astrocytes. As miRNA124-3P is expressed in neurons and EVs derived from them, but not astrocytes, it was used as a marker to assess functional delivery from neurons to astrocytes. GFP-positive astrocytes, resulting from uptake of neuron-derived EVs, were isolated by flow cytometry and found to contain miR-124-3p. GW4869 was given to inhibit EV release. GW4869 inhibits functioning of sphingomyelinase-2 (nSMase2), a key enzyme for converting sphingomyelin to ceramide. Ceramide is a lipid highly abundant in the EV membrane as result of its recruitment during exosome biogenesis, and disruption of this pathway leads to failure of small EV release [25]. When GW4869 was given, a reduction of GFP levels in astrocytes and decreased miR-124-3p levels in GFP^+^ astrocytes was observed. miR-124-3p indirectly upregulates glutamate transporter 1 (GLT1), a protein involved in the clearance of excess glutamate neurotransmitters from the synapse [26]. To inhibit miR-124-3p selectively in astrocytes, miR-124-3p sponges were delivered via an AAV viral vector (AAV5) with an astrocyte-specific promotor (GFAP). miR-124-3p sponges (sp) express sequences complementary to miR-124-3p, thus selectively inhibiting function of this miRNA [27]. This AAV5-37gfap-GFP/miR-124-sp virus was then injected into the motor cortex of adult eaat2-tdT mice. Eaat2-tdT mice express the red fluorescent tdTomato protein in >80% of cortical astrocytes, facilitating downstream analysis of these cells [28]. Both the hippocampus and cortex, GLT1 immunostaining of sections of AAV5-miR-124-sp injected mouse brains revealed a reduction in GLT1 levels only in GFP+ astrocytes. In conclusion, this study demonstrated in vivo that neuron-derived EVs are taken up by astrocytes (Figure 1A) and can change the neurotransmitter profile of these astrocytes. The exact mechanisms and further nuances remain to be elucidated.

### 2.2. Synaptic Development and Signaling

Recent studies in Drosophila have shown that Wingless-related integration site (Wnt) proteins are important in synaptic development, neuroplasticity and signaling. Wnt/Wingless signaling is vital in the development of the nervous system, regulating the formation of the neuronal cytoskeleton and neuromuscular junctions (NMJs), and guiding differentiation of synapses in, for example, the cerebellum [29,30]. As Wnt is hydrophobic, it cannot diffuse in the extracellular space, thus it travels in association with EVs through its binding to the multi-pass transmembrane protein, g-binding protein Evenness Interrupted (Evi) at the NMJ [31] (Figure 1B). Interestingly, electron microscopy of EVs at larval NMJs showed a high number of MVBs that contained Evi at synaptic boutons which ultimately fused with the postsynaptic membrane to deliver their cargo, such as Wnt proteins, to the recipient postsynaptic muscle cells. Downregulating Evi in the presynaptic motor neurons reduced its immunoreactivity at both presynaptic and postsynaptic terminals, suggesting that postsynaptic Evi proteins were transferred via EVs derived from presynaptic terminals [29]. Moreover, fewer synaptic boutons were observed at NMJ derived from larvae with reduced Evi protein levels compared with the control NMJ [31]. The presence of these vesicles containing Evi provides evidence for the essential role of EVs in transferring protein, such as Wnt, to strengthen synaptic connections.

### 2.3. Oligodendrocyte-Derived EVs in the CNS

Oligodendrocytes support neurons by forming myelin sheaths around the axons, which decreases the neuron’s capacitance and increases the velocity of the action potential through the neurons. They also provide trophic support to neurons, sustaining long-term axonal integrity by maintaining the stability of neurons during nutrient-deprived conditions [11]. The role of oligodendrocyte-derived EVs in oligodendrocyte-neuron communication and neuronal functioning is increasingly understood. Recent in vivo studies have underlined some of the biological factors that can trigger the release of EVs from oligodendrocytes, as well as their impact on the myelination of axons [11].

Glutamate, released as a result of depolarization in neurons, can trigger the secretion of EVs from oligodendrocytes by stimulating Ca^2+^ entry through N-methyl-D-aspartate (NMDA) and α-amino-3-hydroxy-5-methl-4-isoxazolepropionic acid (AMPA) receptors [11]. These EVs are subsequently internalized by neurons, which has been shown to improve neuronal viability under cell stress conditions in vitro [32]. In a zebra fish model, oligodendrocyte-derived EVs were shown to affect the extent of myelination of axons [33]. In this experiment, tetanus toxin (TeNT) was used to abrogate the release of EVs from oligodendrocytes to neurons in vivo. Using transmission electron microscopy (TEM) a 39% decrease in the number of myelinated axons in TeNT expressing animals was observed. The onset of myelination was not delayed, but the number of axons myelinated per oligodendrocyte was severely impaired [33].

Oligodendrocyte EVs have been implicated in mediating long-term axonal integrity and maintaining neuronal stability during neurodevelopment [32]. Oligodendrocyte EVs contain Proteolipid Protein (PLP) [32] and 2′,3′-cyclic nucleotide 3′-phosphodiesterase (CNP) which are key components of the myelin sheath. In a CNP^−/−^ and -null mouse model secondary progressive axonal degeneration develops due to lack of myelination. To compare the changes in EV release, oligodendrocytes were isolated from PLP^−/−^ and CNP^−/−^-null mice and EVs from culture supernatant were collected via ultracentrifugation [34]. A significant decrease in EV release from oligodendrocytes in PLP- and CNP-null mice was detected. This result was further confirmed by western blot analysis showing a decrease in various EV marker proteins, such as ALG-2 interacting protein X (Alix), Tumor Susceptibility Gene 101 (Tsg101), Flotillin-1 (Flot1), 70-kDa heat shock proteins (Hsc/Hsp70), PLP, CNP, and Sirtuin-2 (Sirt2) in the EV fraction compared to wild type mice. However, the mechanistic connections between the PLP/CMP proteins, EV release, and neuronal metabolism remains to be fully elucidated.

### 2.4. EVs in Neurodevelopment

In addition to the role of EVs in mediating intercellular communication in the CNS, they also carry proteins that can alter neuronal circuit development. In vivo studies regarding the role of EVs in neurodevelopment are thus far lacking, but there are indications from in vitro work that they may play an important role. Alterations in EVs can result in neurodevelopmental disorders, such as Rett syndrome [35] which is characterized by the loss of acquired motor and language skills, stereotypic movements, autistic features, and both sleep and respiratory abnormalities. Rett syndrome is caused by mutations in an X-linked gene encoding the methyl-CpG-binding protein 2 (MeCP2) [36]. MeCP2 is an essential epigenetic transcriptional regulator required for proper neuronal development [36] and the EVs that carry dysfunctional MECP2 contribute to the deficits in neural circuit development. One study compared EVs released from MECP2 loss-of-function (LOF) donor neural cultures with EVs from isogenic control neuronal cultures via quantitative proteomic analysis [35]. The presence of proteins related to neurogenesis and synaptic development were lowered in the MEPCP2 LOF EVs compared with control EVs. Moreover, EVs derived from isogenic control cells were found to rescue MECP2 LOF neural progenitor cells by increasing cell proliferation, neurogenesis, and synaptogenesis—suggesting an important role for EVs in carrying proteins and other cargo that can affect neuronal cell fate and synaptic development [35].

EVs may also have a role in stimulating neuronal proliferation during neural development t [35]. In vivo cortical neural cell proliferation was evaluated in the granule cell layer (GCL) of the dentate gyrus in postnatal mice (day four) after injection of EVs [35]. The EVs were derived from rat primary neuronal culture media and isolated with ultracentrifugation. One group was injected with 100 µg/mL proteinase K-treated EVs to deplete the surfaces of EVs from exposed proteins, whereas the other group was injected with non-treated EVs. To detect DNA synthesis in proliferating neuronal cells, 5-ethynyl-2′-deoxyuridine (EdU) was injected at the EV injection site. Mice injected with primary neural EVs had significantly greater EdU^+^ cell density in the GCL as compared to pups with proteinase K-treated EVs. This indicates that the proteome on the outside of EVs may have an important role in regulating target cells in vivo.

## 3. Role of Endogenous EVs in Neurodegenerative Disease

EVs are thought to have a dual role in neurodegenerative diseases (Figure 2). In vivo studies have shown that EV release can act as a defense mechanism for brain cells to remove toxic proteins. However, in some conditions the cargo transported out of cells by EVs can then deliver these toxic contents to other cells, spreading the pathology [32,37,38]. Apoptotic bodies derived from neuronal death are not thought to be major players in spreading neurodegeneration, as they are generally rapidly phagocytized by microglia [39,40]. In contrast, EVs have been implicated in neurodegeneration via spread and uptake of toxic proteins [31]. In terms of lipid content, while brain-derived apoptotic bodies and microvesicles are enriched in phosphatidylserine and phosphatidylinositol due to their biogenesis from the plasma membrane, exosomes are created by inward budding of the endosomal membrane and are enriched in ceramide via conversion of sphingomyelin by neutral sphingomyelinase [25,37]. Ceramide was found to be a trigger for exosome budding in MVBs and to be associated with neurotoxic events mediated by EVs in neurodegenerative diseases [25,38].

Protein aggregation is a key feature in many neurodegenerative disorders. For example, α-synuclein (α-syn) accumulates in Parkinson’s disease (PD), Aβ and tau proteins form aggregates in Alzheimer’s disease (AD) and related tauopathies, and transactive response DNA-binding protein of 43 kDa (TDP-43) aggregates in amyotrophic lateral sclerosis (ALS) [41]. Typically, these proteins have nucleation properties which are prone to initiate the process of protein aggregation and contribute to spreading the disease to regions far from the original defect [42,43]. It has been suggested that EVs can contribute to the spreading of these degenerative protein aggregates in several diseases.

### 3.1. Parkinson’s Disease

Loss of dopaminergic neurons in substantia nigra and the presence of α-syn aggregates in Lewy bodies in the brain are key factors in Parkinson’s disease (PD), causing loss of voluntary movements with rigidity, tremors, and bradykinesia in PD patients [44,45]. α-Syn oligomers can propagate through neurons in a prion-like manner, exacerbating neurodegeneration [46]. α-Syn is endocytosed via the ‘Endosomal Sorting Complexes Required for Transport’ (ESCRT) pathway and transported for degradation to MVBs [47]. MVBs evolve into lysosomes, causing degradation of their cargo, or prior to that fuse with the plasma membrane to release their cargo into the extracellular space, then called exosomes [5,48]. In degradative MVBs, charged multivesicular body protein 2b (CHMP2B) plays a pivotal role in targeting α-syn for degradation [49]. Interestingly, reduced levels of CHMP2B were found in a transgenic mouse model of PD and brain samples of patients with PD and dementia with Lewy bodies (DLB) associated with α-syn accumulation [47]. Overexpression of CHMP2B via intracranial injection of a lentivirus vector in a transgenic mouse model restored the levels of CHMP2B, reduced accumulation of α-syn and ameliorated the neurodegenerative pathology [47]. These data suggest that the degradative MVB pathway is disrupted via downregulation of CHMP2B in α-syn diseases, leading instead to exocytosis of α-syn via EVs and subsequent spreading of this protein within the brain. A therapy targeting EVs biogenesis and restoring CHMP2B levels would rescue the spreading of α-syn within the brain.

Another indication of the importance of EVs in neurodegeneration was revealed when EVs isolated by ultracentrifugation from brain tissue of DLB and AD patients were injected intracranially into wild-type mice [50]. Western blot analysis revealed the presence of α-syn in DLB and of Aβ and tau in both the AD and DLB derived EVs. After injection, α-syn accumulation was detected in neurons and astrocytes, a process shown to be mediated by endocytosis [50]. While this data suggests that EVs can deliver pathogenic forms of α-syn to neurons and astrocytes, and have a role in α-syn accumulation, more studies, namely with different EV isolating methods, should be carried out to confirm the presence of pathological proteins in EVs from patients.

### 3.2. Alzheimer’s Disease

Alzheimer’s disease is a neurodegenerative disorder characterized by the accumulation of amyloid plaques in brain. A central pathological feature of AD is the accumulation of Aβ peptides, in the form of senile plaques and the gradual deposition of hyperphosphorylated tau, leading to neuronal dysfunction [51]. Extensive research has shown that several metabolites of the amyloid-B precursor protein (APP) are toxic to neuronal cells. In normal circumstances, APP is known to have a crucial role in synapses formation [52]. It is not yet clear what initiates and propagates this disease, but studies indicate that only the cis isoform of hyperphosphorylated tau can spread in a prion-like manner, with the uptake of pathologic tau triggering the misfolding and aggregation of endogenous tau in healthy cells [53].

The secretion of full-length amyloid-B precursor protein (flAPP) and APP C-terminal fragments (CTFs) via the exosome secretory pathway has been studied in vivo [14]. First, EVs were isolated from frozen brain tissue of mice, dissociating the brain tissue with papain to release EVs into solution. EVs were then purified by differential ultracentrifugation followed by sucrose gradient density centrifugation. No differences in the number of EVs isolated from wild type and transgenic mice overexpressing human APP (Tg2576) were observed. However, four-fold higher levels of APP and APP-processing products (APP-CTFs and Aβ) were found in transgenic mouse EVs indicating that the amount of flAPP and APP CTFs in EVs is proportional to their expression levels in the brain. Moreover, EVs from both the wild-type and transgenic model showed enrichment of APP fragments compared to the full-length protein, suggesting selective loading of APP metabolites into EVs [14]. These findings suggest that brain cells may use EVs as a means of disposal of APP fragments. However, more studies need to be done to validate the process, as well as the quality and possible contaminants of EVs isolated from direct disruption of brain tissue.

EVs can also be a source of toxic proteins for other, healthy, brain cells. In another study, EVs were isolated from brains of either wild type mice or transgenic rTg4510, a mouse model with pronounced tau pathology that contains the human four-repeat tau with the P301L mutation [54,55]. The authors were able to detect the presence of hyperphosphorylated tau protein inside of rTg4510 brain derived-EVs. Moreover, these tau containing-EVs were shown to trigger aggregation of endogenous tau after being taken up by highly sensitive FRET tau biosensor cells that can detect seeding activity in vitro [54]. To evaluate the effects in vivo, rTg4510 derived-EVs carrying the human form of tau were injected in the brains of 3-month-old WT mice. Neuropathological studies confirmed that rTg4510 derived-EVs were able to accelerate murine tau phosphorylation and trigger aggregation of endogenous murine tau into diffusible oligomers, indicating that neurofibrillary tangle (NFT) seeding events had been initiated. Interestingly, total tau levels were unchanged, suggesting the human hyperphosphorylated tau carried by EVs may modify the existing endogenous murine tau [55]. Overall this data suggests that EVs are a platform for triggering pathological modifications in tau proteins and potentially contribute to the development and spreading of tau pathology in AD [54].

Microglia can facilitate tau protein propagation between neurons by phagocytosing and releasing EVs containing tau protein [56]. A rapid tau propagation mouse model showed tau transmission from the entorhinal cortex (EC) to the dentate gyrus (DG) within 28 days after the injection of an adeno-associated virus (AAV) vector expressing a tau transgene under a neuron-specific promoter. In vivo data showed microglia-derived EVs were found to be more efficient in delivering tau to neurons than was the naked protein. In line with this finding, microglia depletion suppressed tau propagation in the dentate gyrus. In addition, EV biogenesis was reduced via administration of GW4869. Administration of GW4869 or an siRNA against nSMase2 suppressed ceramide production and was highly effective in slowing tau propagation within the brain [56]. This in vivo data suggests that microglia-derived EVs are involved in the progression of tauopathy and that targeting EV biogenesis may be a therapeutic approach to alleviate the spread of the disease.

Serum from patients with AD and AD-mimicking mouse models contains “astrosomes”, astrocyte-derived EVs enriched in ceramide, that are associated with the Aβ peptide [38,57]. Serum-derived astrosomes isolated by precipitation are taken up by neural cells and transport Aβ to mitochondria, inducing mitochondrial damage and apoptosis in neurons [38]. In WT mice, intracranially injected astrosomes were taken up by healthy neurons, exposing them to the carried Aβ peptides. The internalization of Aβ peptides promotes caspase activation and leads to mitochondrial damage [58]. These results demonstrate that Aβ accumulation is associated with ceramide in astrosomes and has a substantial role in inducing Aβ neurotoxicity in vitro and in vivo. In line with that idea, administration of GW4869 intraperitoneally in 5XFAD mice, a mouse model of AD with widespread intraneuronal amyloid immunoreactivity, was able to reduce the overall levels of ceramide in the brain and in brain-derived EVs in serum. This led to a significant reduction in the total amyloid plaque area in vivo [57]. These studies suggest that pharmacological inhibition of nSMase2 and EV reduction could be a therapeutic approach for AD.

### 3.3. Amyotrophic Lateral Sclerosis

In contrast to the previous findings, an interesting report in amyotrophic lateral sclerosis and frontotemporal lobar degeneration (FTLD) showed a different response to the administration of GW4869 in vivo [59]. ALS and FTLD are neurodegenerative disorders characterized by ubiquitin-positive inclusions of misfolded proteins, particularly found in glia and neurons. TAR DNA-binding protein of 43 kDa aggregation is the pathological hallmark of ALS and FTLD, and is known to be secreted in EVs [41,60]. After human brain extraction the exosomal TDP-43 was found to be increased in ALS and the levels of full length and C-terminal TDP-43 were found to be upregulated in the EV fraction [59]. Aggregation of TDP-43 in neuronal cells can promote exosomal secretion of TDP-43 species with the release of EVs acting as a protective means to clear the aggregated forms of TDP-43. In vitro data showed that recipient cells were able to take up brain-derived EVs by endocytosis leading to cellular TDP-43 accumulation. However, when EV secretion was inhibited with GW4869 administration to TDP-43^A315T^ transgenic mice, higher levels of TDP-43 aggregation were detected, and denervation of neuromuscular junctions was increased. Abnormalities in the behavior of mice were detected by reduced rotarod performance and memory deficits. Despite exosomal TDP-43 secretion being increased and potentially promoting the propagation of the proteinopathy in human brain, in vivo mice data suggests that EV secretion may actually have an overall beneficial role in TDP-43 clearance. Therapeutic strategies reducing EV release may adversely affect the disease progression in ALS [59].

Overall, in neurodegenerative diseases, in vivo data shows a dual role for EVs. In PD and AD, the data suggests that EVs derived from neurons, astrocytes and microglia can trigger the pathology in healthy cells. Inhibition of EVs secretion, by targeting lipid formation alleviates neuropathological features in these models, being presented as a therapeutic target for the disease. In contrast, the same approach would seem to be disadvantageous in vivo in the context of ALS and FTLD models, potentially exacerbating disease neuropathology and phenotype as observed in mouse models.

## 4. Stroke and Traumatic Brain Injury

Stroke remains one of the most common causes of death and disability in industrialized countries [61]. The data regarding EVs in stroke and trauma is currently sparse, even though these diseases are the most common neurologic diseases. Ischemic stroke occurs when a cerebral artery is occluded by a blood clot or thrombus, usually as a result of chronic atherosclerosis of the vessel wall. The tissue downstream of this blockage becomes ischemic and dies. Over the years more nuance has been brought to the interplay between blood clots and the brain parenchyma, showing important roles for neurons, oligodendrocytes, microglia, astrocytes and immune cells in the mechanisms of tissue injury and repair [62]. EVs are important in the communication between these cells, but the vast majority of EV related research has come from in vitro studies [63]. Several studies have shown that the number of EVs, or EV-like microparticles, is increased after ischemic stroke [64]. These can be derived from neural precursor cells and other cells in the blood and vascular compartment [64]. An increase of EVs carrying pro-inflammatory proteins into the blood has also been observed after stroke [65]. In vivo research has mainly focused on using these released EVs as biomarkers, or in utilizing mesenchymal stromal cell-derived EVs for the treatment of stroke. These developments are reviewed elsewhere [66,67] and beyond the scope of this review.

Similarly, in traumatic brain injury (TBI) and spinal trauma, not much is known about the role of EVs in vivo. Traumatic injury of the brain and spinal cord is among the leading causes of death and disability, especially among young adults [68]. Not only do the location and severity of the initial injury vary wildly among cases, but the disease progression itself is highly heterogeneous, including a mixture of structural damages, hypoxia, hemorrhage, and axonal shearing, followed by various cascades of coagulation, inflammation, and wound healing [69]. The role of EVs in this process is under investigation, focusing primarily on the possibilities of using circulating EVs to diagnose the severity of injury [70]. While EVs released into the plasma after TBI have been widely studied, less is known about the EVs released within the brain itself. One study isolated EVs via sequential filtration and ultracentrifugation from the ispi-and contralateral hemispheres from mice seven days after TBI [71]. RNA in these EVs was sequenced, comparing the hemisphere of trauma, the contralateral hemisphere and a hemisphere from control mice. miR-21 was among the most upregulated miRNAs. Via immunofluorescence they localized high expression of miR-21 to neurons, but not microglia, possibly indicating neurons as a source of miR-21. miR-21 has a wide range of functions and has been implicated in various neuroprotective processes, protecting neurons from ischemic death in stroke [72] and reducing glial scarring in spinal trauma [73].

## 5. Brain Tumors

Brain tumors are known for their intricate relationship with their microenvironment, thriving off subjugation of the surrounding ‘normal’ brain [2,74]. Gliomas and the process of brain metastasis of peripheral cancers are the most studied in this regard, as discussed below (Figure 3A,B).

### 5.1. Glial Tumors

Gliomas are malignant primary CNS tumors with a poor survival, especially in WHO grade IV tumors (glioblastoma-GBM) which constitute the majority of cases [75]. The latter remain impossible to cure due to a litany of factors. One of the major problems is the extensive modulation of the tumor microenvironment which serves to evade an immune response and promote angiogenesis, tumor progression and invasion, as well as resistance to therapy [2]. EVs are known to play an important role in tumor progression [76]. Here we will discuss various mechanisms and processes related to EVs that have been studied in vivo (Figure 3A).

Glioma cells are highly proliferative and can stimulate cell division and potentially even malignant transformation in neighboring cells via transfer of EVs with oncogenic cargos [77,78]. One of these cargos is the chloride intracellular channel-1 (CLIC1), which is present in glioma stem cell (GSC)-derived EVs [79]. CLIC1 is overexpressed in GBM and has a role in promoting cell cycle progression, resistance to pharmaceutical compounds and GSC proliferation [80]. Isolated EVs from GSCs overexpressing CLIC1 increased tumor growth and engraftment when co-injected with a human glioma-derived cell line in an orthotopic xenograft model [79]. Conversely, EVs containing miR-1 have been shown to decrease tumor progression and angiogenesis in mouse models in vivo [81]. Targeting Annexin A2 (ANXA2), a protein previously implicated in GBM invasion and progression [82], miR-1 also decreased the release of EVs by the tumor, in turn leading to decreased progression and angiogenesis.

Glioma-derived EVs are taken up by various normal cells surrounding the tumor. One study illustrated this by using an Ai14 mouse line, which has a loxP flanked STOP cassette preventing transcription of tdTomato. When Cre is introduced in any of the cells in this mouse, tdTomato is expressed and a red fluorescent signal can be detected [83]. These mice were implanted with GFP/Cre-expressing glioma cells [84]. Functional transfer of Cre to non-glioma cells triggered tdTomato expression in these cells, identifying which cells had interacted with glioma EVs. Post-mortem staining of brain tissue showed that the majority of tdTomato-expressing cells were astrocytes (~50%) and endothelial cells (~25%). Interestingly, different glioma cell lines, derived from different precursor cell types showed different affinity for EV uptake by surrounding cells. In neuronal stem cell (NSC) derived gliomas, neurons were responsible for the vast majority of EV uptake (~80%), while for oligodendrocyte precursor (OPC)-derived gliomas uptake was seen both in neurons (~60%) and astrocytes (~20%) [84]. This EV-mediated interaction between astrocytes and glioma cells was further visualized in vivo in a transgenic mouse model with GFAP-driven tdTomato expression in astrocytes in brains implanted with GFP-expressing glioma cells. Intra-vital imaging showed gradual increase of GFP in tdTomato-expressing cells, confirming uptake of glioma-derived EVs by astrocytes [84]. This was not only observed in cells with direct contact to glioma, but also in more distant astrocytes, indicating that this effect was mediated both via direct cell-to-cell contact, and also potentially by EVs. Finally, this study showed changes in synaptic activity between neurons that had taken up glioma-derived EVs versus those which had not, with glioma-EV positive neurons having a higher frequency of spontaneous synaptic response, which has been related to increased tumor growth [85]. This study uniquely showed the variability in uptake of GBM-derived EVs in vivo by different cell types, with the primary origin of the tumor playing a major role. Different tumor types may succeed by modulating specific non-tumor cells, necessitating targeted approaches when developing treatments aiming to interfere in this process.

Glioblastomas are known for their high demand for nutrients and oxygen, driving angiogenesis in the surrounding area [86]. Glioblastoma-derived EVs contain many angiogenic factors, such as miR-26a, miR-21a, vascular endothelial growth factor (VEGF), and long non-coding RNA Brain-1 [87,88,89,90]. EVs released by hypoxic glioma cells may be especially adept at inducing an angiogenic response. In a xenograft model of GBM, injecting a human derived glioma cell line together with EVs derived from glioma cells grown under hypoxic conditions accelerated tumor growth and strongly enhanced neovascularization [91]. Another xenograft study showed that EVs from hypoxic cells could facilitate resistance to radiotherapy via miR-301a-mediated downregulation of Transcription Elongation Factor A Like 7 (TCEAL7) [92]. As these studies were done by injecting human derived glioma cells subcutaneously in immune compromised mice, the real relationship between EVs from hypoxic cells, angiogenesis, and gliomas in the brain remains to be further investigated.

Glioblastoma modulate the immune system via EVs [93]. To study this interaction, one group utilized CX3CR1^GFP/+^ mice [94]. CX3CR1 is expressed on microglia, macrophages, and monocytes, and this mouse model GFP has been fused to CX3CR1, labeling these cells green [95]. These mice were injected with a glioma cell line expressing palmitoylated mCherry, which labels glioma-derived EVs red. As such, uptake of glioma-derived EVs by microglia, monocytes, and macrophages could be visualized via intravital imaging. Functional uptake of these EVs was later confirmed in microglia by changes in their transcriptome profile [93]. Tumor cells (with high miR-21 content) expressing palmitoylated GFP were implanted in mir-21-null mice. Microglia were then isolated based on the GFP signal, indicating uptake of glioma-derived EVs. Sequencing showed an increase in miR-21 in these microglia and also significant downregulation of some mir-21 target mRNAs. This change was not seen in GFP-negative microglia from the same brains, indicating EVs mediate functional transfer of miR-21 from glioma cells to microglia in their vicinity. Another study showed that intracranial injection of glioma-derived EVs in a mouse model of glioma significantly reduced the infiltration of CD8-positive T cells, known to be important in anti-tumor immunity [96]. Together, these studies confirm that GBM modulates the immune system via EVs in vivo. Not only are the EV cargos important in this process, but the expression of proteins on the outside of the EVs, as has been shown with Programmed cell Death protein 1 (PD-1) [77], also have a role. EVs are only one facet of immunosuppression by GBM, but one that will hopefully lend itself to specific targeting in future treatments. Studying the effects of inhibiting EV release in glioma has however had mixed results. Knockdown of Ras-associated protein 27a (Rab27a), a small GTPase, in glioma cells decreased release of small EVs in vitro and slowed tumor growth in vivo [84,97]. However, as the function of Rab27a in glioma is multifaceted, knocking down Rab27a may have unwarranted downstream effects aside from inhibition of EV release, such as changes in CCL2 expression, and should therefore be used with caution in glioma [97].

### 5.2. Brain Metastasis

Brain metastasis are the most common neurologic complication in cancer, with lung cancer, breast cancer, and melanoma the most likely tumors of origin [98]. Metastasis in cancer is a highly complex process, with brain metastasis being poorly understood [74,99]. EVs appear to have a role in facilitating metastasis to the brain (Figure 3B). EVs derived from breast cancer cell lines with high brain metastatic potential were shown to increase the permeability of the BBB, and easily entered the brains of mice from the vasculature as compared with EVs isolated from low-metastatic breast cancer cells [100]. Pre-treatment with EVs derived from brain-metastatic breast cancer cell lines injected intravascularly increased the frequency of brain metastasis by cardiac injected breast cancer cells in two separate studies [100,101]. In vitro work has suggested that miR-181c, found in these brain metastatic EVs, may facilitate brain metastasis via downregulation of 3-phosphoinositide-dependent protein kinase-1 (PDPK1), which is critical in maintaining tight junctions in the BBB [100]. Similarly, miR-105 was also found in these EVs, with EVs from cells with overexpressed miR-105 inducing destruction of tight junctions—targeting downregulation of zonula occludens protein-1 (ZO-1) [101]. Finally, long non-coding RNA GS1-600G8.5 was found to be upregulated in EVs derived from metastatic breast cancer, potentially contributing to BBB disruption via downregulation of BBB proteins ZO-1, claudin-5, and N-cadherin [102].

Tumor-derived EVs are also able to change glucose metabolism in the brain via miR-122 [103]. Intravenously injected breast cancer-derived EVs with high miR-122 levels could be found in astrocytes and were associated with reduced glucose uptake in the brain. Downregulation of pyruvate kinase, an enzyme important for glycolysis in cancer cells, was also observed. Pre-treatment of the brain by IV injection of EVs having high levels of miR-122 significantly increased metastatic colonization of breast cancer cells in the brain. Although no experiment was performed with inhibition of EV release from these tumors, this study demonstrates the potential of breast cancer-derived EVs to prime the brain for engraftment of circulating tumor cells [103].

As with glial tumors, brain metastases interact with cells in their microenvironments, drawing support from ‘normal’ surrounding cells. One study demonstrated the interaction between astrocytes and metastatic breast cancer [104]. In patient data they observed significant downregulation of PTEN (Phosphatase and Tensin Homolog deleted on Chromosome 10), an important tumor suppressor gene, in brain metastases of breast cancer versus non-metastasized tumors. However, knockdown of PTEN in the primary tumor did not result in higher metastatic rates in their in vivo breast cancer model, and low expression of PTEN was not related to increased brain metastases in patient data. They found that astrocytes release miR-19a via EVs, which downregulates PTEN in breast cancer cells. To assess the function of miR-19 in vivo, the Mirc1tm1.1Tyj/J mouse model was used. These mice have a floxable allele encoding for miR-17, miR-18, miR-19a, miR-20a, miR-19b-1, and miR-92-1 [105]. When a Cre recombinase protein is introduced, the allele gets removed and expression is diminished. To silence miR-19 in astrocytes, an AAV-GFAP-Cre virus was injected IV. This significantly suppressed formation of brain metastases, indicating the importance of this cluster of microRNAs in the formation of brain metastases. This model is however not specific for miR-19, so further research is warranted. The role of EVs was confirmed by knockdown of Rab27a/b via a short hairpin RNA lentivirus in the brain parenchyma, which decreased the release of EVs, reduced the down regulation of PTEN and subsequently decreased metastases and tumor growth in the brain. This effect could then be rescued by co-injecting tumor cells with astrocyte-derived EVs into the brain leading to significant PTEN downregulation and tumor outgrowth. Overall, this study demonstrated the intricate relationship between astrocytes and brain metastases mediated by astrocyte-derived EVs. These studies demonstrated that distant tumors can prime the brain to form premetastatic niches via release of EVs, and suggest that resident brain cells may actually facilitate metastases in certain malignancies.

## 6. Conclusions and Future Prospects

The last few years has seen major developments in studies evaluating the role of EVs in the brain in vivo. Driven by developments in imaging techniques and transgenic mouse models, many concepts developed in vitro can now be evaluated in vivo. With availability of these methods, more caution should be placed in drawing conclusions from in vitro work with EVs. Rather EVs should be studied in the context of the various intricate cell communication processes in normal and diseased brain tissue. Creating a deeper, more complete understanding of the role of EVs in these processes will allow a more nuanced understanding of molecules and mechanisms involved, as well as improving the chances of translating these findings from the laboratory to the clinic.

## 7. Material and Methods

Figures created with BioRender.com.

## Figures and Tables

**Figure 1 ijms-22-07294-f001:**
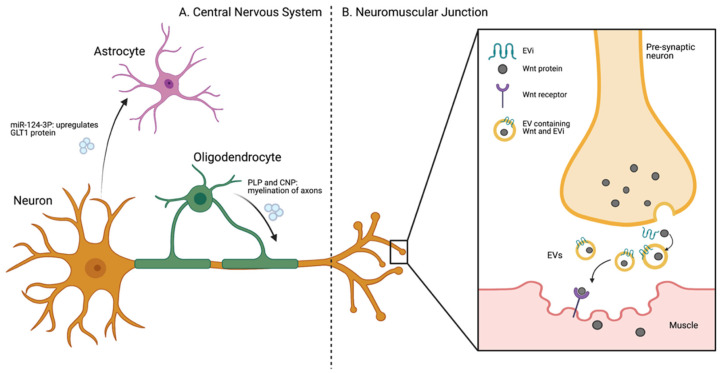
Schematic overview of in vivo data regarding extracellular vesicles in the brain and neuromuscular junction. (**A**) Overview of EV-related communication in the Central Nervous System (CNS). In vivo data has shown that neuron-derived EVs are taken up by astrocytes, changing protein expression in these cells. Oligodendrocyte-derived EVs have been linked to myelination of axons. CNP: 2′,3′-cyclic nucleotide 3′-phosphodiesterase, GLT1: glutamate transporter 1, PLP: proteolipid protein. (**B**) The role of EVs in signaling at the neuromuscular junction. EVi: Evenness Interrupted, EVs: extracellular vesicles, GLT1: glutamate transporter 1, Wnt: Wingless-related integration site.

**Figure 2 ijms-22-07294-f002:**
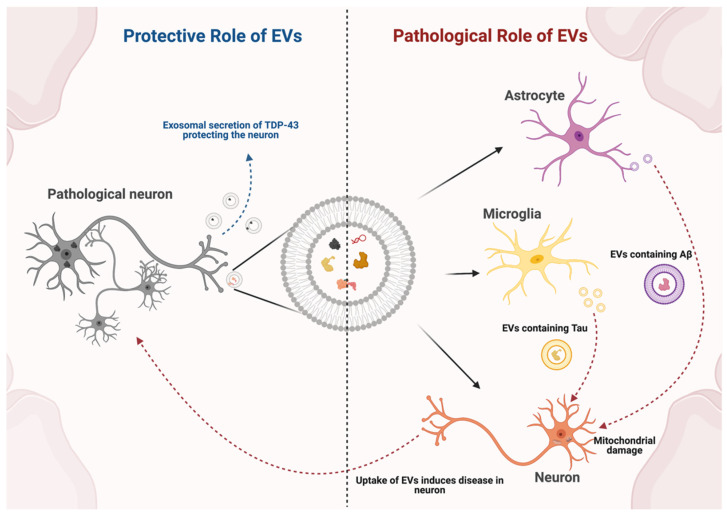
Dual role of endogenous extracellular vesicles in neurodegenerative disease in vivo. On the left, neurons use EVs to dispose of TPD-43, a toxic protein implicated in amyotrophic lateral sclerosis (ALS). On the right, toxic proteins are released from astrocytes and microglia in Alzheimer’s and Parkinson’s disease, spreading toxic proteins to neurons. EV: extracellular vesicle.

**Figure 3 ijms-22-07294-f003:**
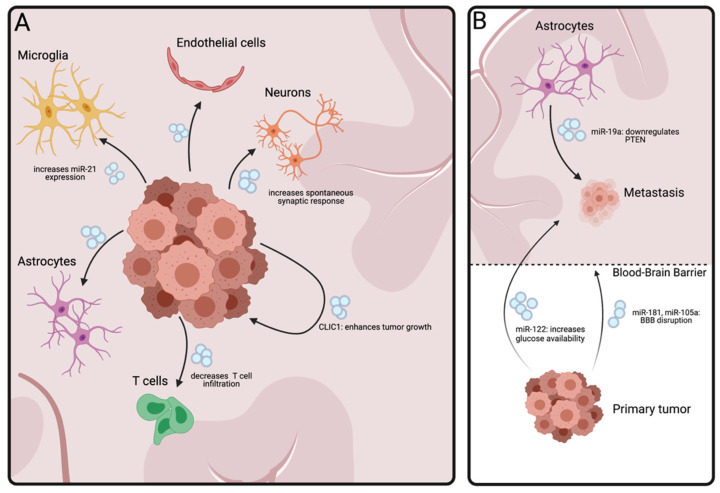
Communication between tumors and the brain. (**A**) Extracellular Vesicle mediated interaction between a glioma and the tumor microenvironment. A glioma releases EVs into its microenvironment, interacting with the surrounding cells. Glioma-derived EVs also provide a positive feedback loop between tumor cells, transferring oncogenic drivers. (**B**) EV-derived communication between brain metastasis, distant tumors and the brain. Primary tumor releases EVs that can disrupt the blood–brain barrier, or prime the brain for metastasis. In the brain, astrocyte-derived EVs can promote metastasis growth via release of PTEN. BBB: Blood–brain barrier; CLIC-1: Chloride Intracellular Channel-1; PTEN: Phosphatase and Tensin Homolog deleted on Chromosome 10.

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
