# Peer review of "Living Proof of Activity of Extracellular Vesicles in the Central Nervous System"

_ijms, 2021, doi:10.3390/ijms22147294_

Round 1
Reviewer 1 Report
An exciting review summarizes what is known so far about the role of EVs in the proper functioning of the central nervous system. The work also deals with the role of EVs in the physiopathology of the nervous system.
Please find below my comments:
1. the title of the manuscript - it is fascinating and encourages the reader to read the manuscript,
2. the remaining chapters are detailed, and I do not bring negative comments to them,
3. a critical remark concerns the description of the role of EVs in the pathophysiology of stroke. The authors should describe their role in ischemic and hemorrhagic stroke. I also suggest using the publication by Dr. Chiva-Blanch and Dr. Switonska on the role of MPs in ischemic stroke.
Author Response
Reviewer 1:
An exciting review summarizes what is known so far about the role of EVs in the proper functioning of the central nervous system. The work also deals with the role of EVs in the physiopathology of the nervous system.
Please find below my comments:
- the title of the manuscript - it is fascinating and encourages the reader to read the manuscript,
- the remaining chapters are detailed, and I do not bring negative comments to them,
- a critical remark concerns the description of the role of EVs in the pathophysiology of stroke. The authors should describe their role in ischemic and hemorrhagic stroke. I also suggest using the publication by Dr. Chiva-Blanch and Dr. Switonska on the role of MPs in ischemic stroke.
We thank the reviewer for the kind comments regarding our manuscript. We found the paper suggested to be of great interest and have added it to our review. Page 9, line 362-6:
“Several studies have shown that the number of EVs, or EV-like microparticles, is increased after ischemic stroke[66]. These can be derived from neural precursor cells and other cells in the blood and vascular compartment[66].”
Reviewer 2 Report
Manuscript concerns an important issue of the role of extracellular vesicles in health and pathology of CNS as evaluated by in vivo investigations. Particular important is revealing and pointing to the dual action of various type of EV in different pathologies of CNS and evaluation of their application in the therapy. Review is well written, clear and easy to follow with citations of actual literature. For better assimiliation the very minor supplementations and/or explanations are suggested concerning following points:
2.1. Neuron-Astrocyte Communications:
Line 93: GW4869 : no explanation of the mechanism of its action. Actually it is given in parts Alzheimer’s Disease and following, but should be in the first use of that abrevation.
Line 97:Additional explanation will be very useful concerning the use of the model of the „miR- 97 124-3p sponge (sp) was applied using an AAV5-gfap-GFP/miR-124-sponge” , „AV5-37gfap-GFP/miR-124-sp virus”and „eaat2-tdT mice, in which the tdT reporter is selectively expressed „
Also no comments are given to Figure 1. Part A) Overview of EV-related communication in the Central Nervous System (CNS).
2.3. Oligodendrocyte-derived EVs in the CNS: Line 133: is „speed of signal transduction. decreases the capacitance of the neuron” should be rather : decreases the capacitance, and increases the velocity of the action potential of the neurons
Line 160: it would be nice to include of the explanation of following abreviations”: Alix, Tsg101, Flotillin-1 (Flot1), Hsc/Hsp70, PLP, CNP”
3.3. Amyotrophic lateral sclerosis: Line 339: „This suggests that EV secretion may actually have a beneficial role in TDP-43 clearance and therapeutic strategies reducing EV release may adversely affect the disease progression in ALS „ What about accumulation of TDP-43 in oother recipient cells in the brainundr that conditions?
5.1. Glial tumors:
Few more words of explanation concerning the used model and abreviations will be useful: Line 418: „Ai14 mouse line, which has a loxP flanked STOP 418 cassette preventing transcription of tdTomato, implanted with GFP/Cre-expressing glioma cells. Functional transfer of Cre to non-glioma cells triggered tdTomato expression in these cells”
The same concerns: Line 455: „ CX3CR1GFP/+ mice, 455 which labels microglia, macrophages, and monocytes with GFP, and a glioma cell line 456 expressing palmitoylated Cherry” and abreviations : PD-1, Rab27a/b.
5.2. Brain metastasis :
Line 501: pyruvate kinase: its isoenzyme in glycolysis of cancer cells
Line 515 : a few more words on model „AAV-GFAP-Cre 515 in Mirc1tm1.1Tyj/J mice”
Author Response
Reviewer 2:
Manuscript concerns an important issue of the role of extracellular vesicles in health and pathology of CNS as evaluated by in vivo investigations. Particular important is revealing and pointing to the dual action of various type of EV in different pathologies of CNS and evaluation of their application in the therapy. Review is well written, clear and easy to follow with citations of actual literature. For better assimilation the very minor supplementations and/or explanations are suggested concerning following points:
We thank the reviewer for the kind words and appreciate the extensive comments. We have detailed our responses below in red.
2.1. Neuron-Astrocyte Communications:
Line 93: GW4869 : no explanation of the mechanism of its action. Actually it is given in parts Alzheimer’s Disease and following, but should be in the first use of that abrevation.
We thank the reviewer for pointing this out. We have moved this explanation to the first mention of GW4869, line 93:
“GW4869 was given to inhibit EV release. GW4869 inhibits functioning of sphingomye-linase-2 (nSMase2), a key enzyme for converting sphingomyelin to ceramide. Ceramide is a lipid highly abundant in the EV membrane as result of its recruitment during exosome biogenesis, and disruption of this pathway leads to failure of small EV release[25].”
Line 97:Additional explanation will be very useful concerning the use of the model of the „miR- 97 124-3p sponge (sp) was applied using an AAV5-gfap-GFP/miR-124-sponge” , „AV5-37gfap-GFP/miR-124-sp virus”and „eaat2-tdT mice, in which the tdT reporter is selectively expressed „
We have improved our explanation of this model on page 3, line 100-6.
“To inhibit miR-124-3p selectively in astrocytes, miR-124-3p sponges were delivered via an AAV viral vector (AAV5) with an astrocyte-specific promotor (GFAP). miR-124-3p sponges (sp) express sequences complementary to miR-124-3p, thus selectively inhibiting function of this miRNA[27]. This AAV5-37gfap-GFP/miR-124-sp virus was then injected into the motor cortex of adult eaat2-tdT mice. Eaat2-tdT mice express the red fluorescent tdTomato protein in >80% of cortical astrocytes, facilitating downstream analysis of these cells[28]. “
Also no comments are given to Figure 1. Part A) Overview of EV-related communication in the Central Nervous System (CNS).
We now refer to the figure as a number in the text (i.e. line 86 and 110) and have expanded the legend.
2.3. Oligodendrocyte-derived EVs in the CNS: Line 133: is „speed of signal transduction. decreases the capacitance of the neuron” should be rather : decreases the capacitance,and increases the velocity of the action potential of the neurons
We thank the reviewer for pointing this out. We have added this to our paper, page 4 line 141-3.
“Oligodendrocytes support neurons by forming myelin sheaths around the axons, which decreases the neuron’s capacitance and increases the velocity of the action potential through the neurons.”
Line 160: it would be nice to include of the explanation of following abreviations”: Alix, Tsg101, Flotillin-1 (Flot1), Hsc/Hsp70, PLP, CNP”
We had provided the explanations for PLP and CNP on page 4, line 161-3 and have now changed the text to explain the other abbreviations as well. The text now reads:
“Oligodendrocyte EVs contain Proteolipid Protein (PLP)[33] and 2′,3′-cyclic nucleotide 3′-phosphodiesterase (CNP) which are key components of the myelin sheath.”
We have added explanations of the other abbreviations on page 5, line 168-71.
“This result was further confirmed by western blot analysis showing a decrease in various EV marker proteins, such as ALG-2 interacting protein X (Alix), Tumor Susceptibility Gene 101 (Tsg101), Flotillin-1 (Flot1), 70-kDa heat shock proteins (Hsc/Hsp70), PLP, CNP, and Sirtuin-2 (Sirt2) in the EV fraction compared to wild type mice.”
3.3. Amyotrophic lateral sclerosis: Line 339: „This suggests that EV secretion may actually have a beneficial role in TDP-43 clearance and therapeutic strategies reducing EV release may adversely affect the disease progression in ALS „ What about accumulation of TDP-43 in oother recipient cells in the brainundr that conditions?
We agree with the reviewer that it would be highly interesting to explore accumulation of TDP-43 in other cells. Unfortunately, the authors of the cited paper did not explore in mice the accumulation of exosomal TDP43 in other brain recipient cells. The evidence they presented in mice was based in the aggregation of TDP43 in brain progenitor cells and the phenotypic impact of blocking EVs release in mice. We reformulated the sentence on page 8, line 347-51:
“Despite exosomal TDP-43 secretion being increased and potentially promoting the propagation of the proteinopathy in human brain, in vivo mice data suggests that EV secretion may actually have an overall beneficial role in TDP-43 clearance. Therapeutic strategies reducing EV release may adversely affect the disease progression in ALS[63].”
5.1. Glial tumors:
Few more words of explanation concerning the used model and abreviations will be useful: Line 418: „Ai14 mouse line, which has a loxP flanked STOP 418 cassette preventing transcription of tdTomato, implanted with GFP/Cre-expressing glioma cells. Functional transfer of Cre to non-glioma cells triggered tdTomato expression in these cells”
We have added a brief explanation of the mouse model and performed experiment. Page 10 line 431-6.
“One study illustrated this by using an Ai14 mouse line, which has a loxP flanked STOP cassette preventing transcription of tdTomato. When Cre is introduced in any of the cells in this mouse, tdTomato is expressed and a red fluorescent signal can be detected[87]. These mice were implanted with GFP/Cre-expressing glioma cells[88]. Functional transfer of Cre to non-glioma cells triggered tdTomato expression in these cells, identifying which cells had interacted with glioma EVs.”
The same concerns: Line 455: „ CX3CR1GFP/+ mice, 455 which labels microglia, macrophages, and monocytes with GFP, and a glioma cell line 456 expressing palmitoylated Cherry” and abreviations : PD-1, Rab27a/b.
We improved our explanation of these models as well. See page 11, line 470-75:
“To study this interaction, one group utilized CX3CR1GFP/+ mice[98]. CX3CR1 is expressed on microglia, macrophages, and monocytes, and this mouse model GFP has been fused to CX3CR1, labeling these cells green[99]. These mice were injected with a glioma cell line expressing palmitoylated mCherry, which labels glioma-derived EVs red. As such, uptake of glioma-derived EVs by microglia, monocytes, and macrophages could be visualized via intravital imaging.”
We have also adjusted page 11, line 488: “as has been shown with Programmed cell Death protein 1 (PD-1)[81],”
We have added a short description regarding Rab27a on page 12, line 491. “Knockdown of Ras-associated protein 27a (Rab27a), a small GTPase”
5.2. Brain metastasis :
Line 501: pyruvate kinase: its isoenzyme in glycolysis of cancer cells
We have added this on page 13, line 20: “Downregulation of pyruvate kinase, an enzyme important for glycolysis in cancer cells, was also observed.”
Line 515 : a few more words on model „AAV-GFAP-Cre 515 in Mirc1tm1.1Tyj/J mice”
We now explain this model in detail on page 13, line 535-542.
“To assess the function of miR-19 in vivo, the Mirc1tm1.1Tyj/J mouse model was used. These mice have a floxable allele encoding for miR-17, miR-18, miR-19a, miR-20a, miR-19b-1 and miR-92-1[109]. When a Cre recombinase protein is introduced, this allele is removed and expression is diminished. To silence miR-19 in astrocytes, an AAV-GFAP-Cre virus was injected IV. This significantly suppressed formation of brain metastases, indicating the importance of this cluster of microRNAs in the formation of brain metastases. This model is however not specific for miR-19, so further research is warranted.”
Reviewer 3 Report
The manuscript entitled "Living proof of activity of extracellular vesicles in the central nervous system" describes the present situation on the roles of EVs in the CNS in vitro and in vivo. I think the review is well organized, and the roles of EVs in the CNS are clearly listed and shown.
I suggest only small minor point,
- The numbering of Figure has some error.
- In references, is there any particular reason you marked some with an asterisk? See references, 10 (in addition the square). 14.
Author Response
We thank the reviewer for reading our manuscript and providing feedback.
- We thank the reviewer for pointing the error in the figures. We double checked all figures and legends, and made some small corrections.
- There were indeed some mistakes in the references. We corrected reference 10 and 14, and went through all other to check for mistakes.
Round 2
Reviewer 1 Report
The authors have addressed all the comments of the reviewers and revised the manuscript accordingly.